# The Impact of Hip Dysplasia on CAM Impingement

**DOI:** 10.3390/jpm12071129

**Published:** 2022-07-12

**Authors:** Carsten Y. W. Heimer, Chia H. Wu, Carsten Perka, Sebastian Hardt, Friedemann Göhler, Tobias Winkler, Henrik C. Bäcker

**Affiliations:** 1Centrum für Muskuloskeletale Chirurgie, Charité—Universitätsmedizin Berlin, Augustenburger Platz 1, 13353 Berlin, Germany; carsten.heimer@charite.de (C.Y.W.H.); carsten.perka@charite.de (C.P.); sebastian.hardt@charite.de (S.H.); tobias.winkler@charite.de (T.W.); 2Department of Orthopaedics & Sports Medicine, Baylor College of Medicine Medical Centre, Houston, TX 77030, USA; wu.h.chia@gmail.com; 3Department of Radiology, Charité Berlin, University Hospital, Chariteplatz 1, 10117 Berlin, Germany; friedemann.goehler@charite.de; 4Julius Wolff Institute, Berlin Institute of Health at Charité—Universitätsmedizin Berlin, Augustenburger Platz 1, 13353 Berlin, Germany; 5Berlin Institute of Health Center for Regenerative Therapies, Berlin Institute of Health at Charité—Universitätsmedizin Berlin, Augustenburger Platz 1, 13353 Berlin, Germany

**Keywords:** radiography, CAM, femoroacetabular, impingement, FAI, PAO, dysplasia, borderline

## Abstract

Predisposing factors for CAM-type femoroacetabular impingement (FAI) include acetabular protrusion and retroversion; however, nothing is known regarding development in dysplastic hips. The purpose of this study was to determine the correlation between CAM-type FAI and developmental dysplastic hips diagnosed using X-ray and rotational computed tomography. In this retrospective study, 52 symptomatic hips were included, with a mean age of 28.8 ± 7.6 years. The inclusion criteria consisted of consecutive patients who suffered from symptomatic dysplastic or borderline dysplastic hips and underwent a clinical examination, conventional radiographs and rotational computed tomography. Demographics, standard measurements and the rotational alignments were recorded and analyzed between the CAM and nonCAM groups. Among the 52 patients, 19 presented with CAM impingement, whereas, in 33 patients, no signs of CAM impingement were noticed. For demographics, no significant differences between the two groups were identified. On conventional radiography, the acetabular hip index as well as the CE angle for the development of CAM impingement were significantly different compared to the nonCAM group with a CE angle of 21.0° ± 5.4° vs. 23.7° ± 5.8° (*p* = 0.050) and an acetabular hip index of 25.6 ± 5.7 vs. 21.9 ± 7.3 (*p* = 0.031), respectively. Furthermore, a crossing over sign was observed to be more common in the nonCAM group, which is contradictory to the current literature. For rotational alignment, no significant differences were observed. In dysplastic hips, the CAM-type FAI correlated to a lower CE angle and a higher acetabular hip index. In contrast to the current literature, no significant correlations to the torsional alignment or to crossing over signs were observed.

## 1. Introduction

The femoroacetabular impingement (FAI) pathology is not only correlated with hip pain but also predisposing for early onset osteoarthritis [1]. It results from an aspherical head–femoral neck junction (CAM type, Figure 1), which is often referred to as pistol grip or post-slip deformity, which typically causes shear stress at the labrum, and cartilage is typically in the anterosuperior region of the acetabulum (pincer type) [2]. These stresses are thought to separate labrum and cartilage, leading to articular degeneration and osteoarthritis [3,4]. Typical causes include acetabular protrusion or acetabular retroversion with anterior overcoverage of the femoral head [1,5]. Thus, especially acetabular retroversion correlated with the development of extra-articular subspace impingement, however, the location of impingement may differ. In addition to underlying biomechanical pathologies of the hip, the range of motion is thought to be important for the development of FAI, also causing an increased shear stress on the labrum and, subsequently, the cartilage [1].

In contrast to CAM-type impingement, the pincer type is not related to an asphericity of the femoral head. Typically, the cause is a deep socket, which limits the hip’s range of motions related to an overcovering acetabular rim. Thus, the femoral neck abuts against the labrum, which is compressed. The forces are transmitted to the acetabular cartilage causing ossification [4].

The opposite of a larger femoral head coverage, such as protrusio acetabula, is developmental dysplasia of the hip (DDH). Typically, it is screened in infants to initiate treatment as early as possible and to allow for a good development of the acetabulum to avoid early onset of secondary hip osteoarthritis [6]. In adults, for diagnosis of DDH, a conventional radiography is performed, showing a low center edge angle, acetabular hip index and acetabular hip angle (AIA). Currently, little is known about the correlating rotational alignment [7] or the presence of CAM FAI in such cases. As a result, surgeons indicate periacetabular osteotomy in DDH without considering or approaching CAM FAIs. Even worse, if the acetabulum is repositioned, a larger femoral coverage can be achieved, which may worsen the development of CAM FAI and, therefore, secondary osteoarthritis.

Because of the dearth in the literature, this study aimed to investigate the radiographic correlation and underlying cause between CAM-type FAI and dysplastic respectively borderline dysplastic hips including the standard measurements of the hip and the rotational alignment of the lower extremity.

## 2. Materials and Methods

A retrospective chart review was performed between 2017 and 2019 after obtaining ethical approval (EA4/201/19). In the period of interest, all consecutive patients aged of 18 years or older presenting with dysplastic (type 2) and/or borderline dysplastic hips (type 1), both types defined by the CE angle, the sharp angle, the acetabular index angle or the presence of a crossing over sign, who underwent a rotational CT scan and two conventional radiographies were included. Further, demographics including age, gender, body weight, body height, body mass index (BMI) and comorbidities were noted. The exclusion criteria consisted of patients without pathological radiographical values, incomplete medical or radiographic charts, no accessible CT images and patients younger than 18 years of age.

The X-rays were performed anteroposteriorly for the pelvis as well as in an axial view and faux profile of the affected hip. In addition to the presence of a CAM femoroacetabular impingement, defined as an extra bone formation at the anterolateral head–neck junction leading to a non-spherical morphology of the femoral head, we analyzed standard measurements including center edge angle (CE), acetabular index angle (AI angle), sharp angle, hip lateralization index, acetabular hip index (AHI) and centrum-collum-diaphyseal angle (CCD) from an anteroposterior view as well as the alpha—vertical line parallel to either the outer and inner cortex of the ilium and the acetabular rim—and beta angles—vertical line parallel to either the outer or inner cortex of the ilium and the lowest and lateral most point of the bright spot of the lower limb of the os ilium—from an axial view. Furthermore, we recorded the presence of pincer-type impingement.

For rotational CT, nonenhanced CTs of the lower limb were obtained either on a 320-row or an 80-row CT scanner (Canon Aquilion ONE Vision Edition and Canon Aquilion PRIME, respectively, both Canon Medical Systems, Tochigi, Japan). Thus, a scanogram and a helical acquisition of the lower limb were obtained, and the scan was performed with 120 KVp tube voltage. An automated tube current modulation was set to the low-dose mode (standard deviation of 25). For image processing, iterative reconstruction (adaptive iterative dose reduction (AIDR) 3D standard) and a bone kernel (filter convolution (FC) 08-H) was used with CT images of 0.5 to 1.0 mm in thick slices.

Measurements were performed on the axial views of the lower limb scan and included the acetabular rotation, defined as the angle between the level of the tangent along the posterior and the anterior acetabular edge and a tangent along the right and left sciatic spina. Additionally, femoral torsion, tibial torsion and tibiofemoral torsion were calculated based on the measurements of the femoral neck (angle between a line through femoral neck and femoral head center and image base line), femoral condyle (angle between tangent along the posterior condyle border and image base line), tibial plateau (angle between a tangent along the posterior edge of the tibial plateau and image base line) and upper ankle rotation (angle between a line through the talus and the lateral malleolus and the image base line). Finally, the tibiofemoral rotation was calculated as the difference between the rotation of the femoral condyles and the rotation of the tibial plateau. All measurements were performed by a musculoskeletal trained radiologist.

The severity of DDH was classified into dysplastic and borderline dysplastic hips as published by Tannast et al. [8,9,10,11]. Therefore, a borderline dysplastic hip was defined as a CE angle between 20° and 24.9°, a sharp angle between 39 and 42° or combined acetabular retroversion (presence of a crossing over sign) with normal values. A dysplastic hip was defined as a CE angle less than 20°, an AI angle greater than 10°, a sharp angle greater than 42° or an AHI greater than 25° [8,9]. The physiological values for acetabular as well as femoral torsions were defined to be between 10° and 25° [12,13]. Two examples are shown in Figure 1 and Figure 2.

A functional clinical examination was performed when the patients first presented in clinics. Thus, the range of motion, including flexion/extension, internal/external rotation and abduction/adduction, were assessed.

For statistical analysis, we used the IBM SPSS Statistics 26 Core System (IBM, Armonk, NY, USA). An ANOVA *t*-test and a mixed model were applied, because they encounter the dependent variable of the person. Additionally, a linear regression analysis was performed to identify cross-correlation significances. Normally distributed continuous variables are presented with the mean and standard deviation of the mean (SD). The level of significances was set to a (*) *p*-value ≤ 0.05.

**Figure 1 jpm-12-01129-f001:**
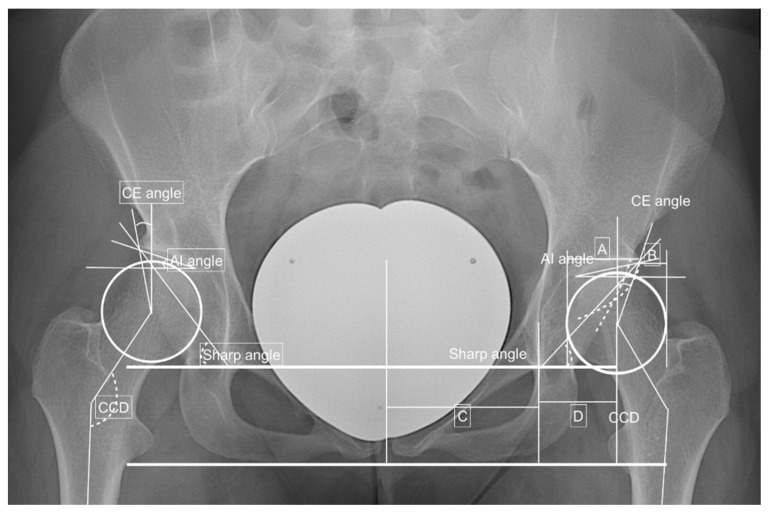
Preoperative findings and measurements performed. Left side: CE angle—between the Perkin line and the line between the center of the femoral head to the lateral edge of the acetabulum −18.8°; AI angle—between the Hilgenreiner’s line and a parallel line to the acetabular roof—12.3°; sharp angle—between the horizontal teardrop line and a line connecting the teardrop to the lateral acetabulum—43.9°; hip lateralization index—quotient between the horizontal distance of the lateral femoral head that is uncovered by the acetabulum divided by a horizontal width of the femoral head—0.54; anterior hip index—quotient of the femoral head that is covered by the acetabular roof and the width of the femoral head—73.9; CCD angle—between the axis of the femoral diaphysis and the axis of the femoral neck—137.4° and positive crossing over sign. Adopted from [14,15].

**Figure 2 jpm-12-01129-f002:**
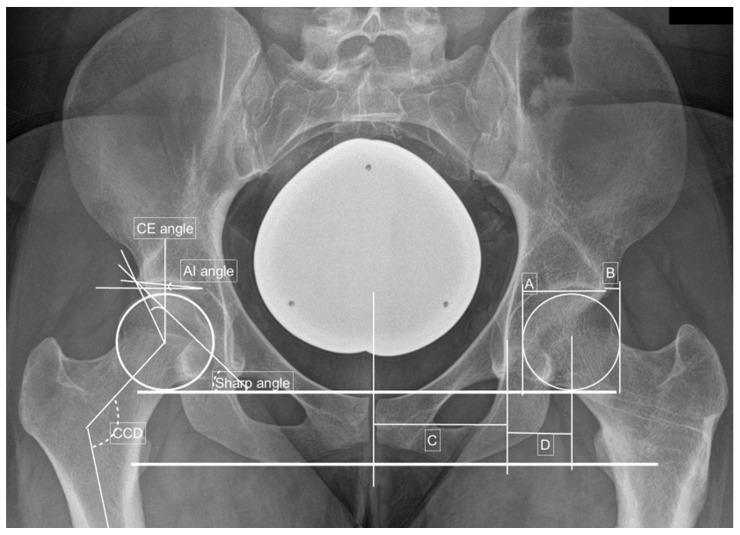
Preoperative measurements in a right-sided borderline dysplastic hip: CE angle—23.4°; AIA—4.9°; sharp angle—40.0°; hip lateralization—0.51; anterior hip index—82.8; CCD angle—121.0°. Adopted from [14,15].

## 3. Results

A total of 52 symptomatic hips of patients at a mean age of 28.8 ± 7.6 years were included. In nineteen of the patients, pathoanatomical parameters of a CAM impingement were observed, whereas in 33 patients, no CAM impingement was noticed. Female patients were predominant, representing 88.5% of cases (n = 46/52). The average height, body weight and body mass index (BMI) were 169.7 ± 8.4 cm, 68.1 ± 12.2 kg and 23.8 ± 4.3 kg/m^2^. No significant differences between the two groups were identified as illustrated in Table 1.

In the functional clinical examination, no significant differences between the groups were observed for internal/external rotation, flexion/extension and abduction/adduction. All findings are presented in Table 2.

For conventional radiography, significant differences between the CAM and nonCAM group were found for the CE angle as well as the acetabular hip index at 21.0 ± 5.4 vs. 23.7 ± 5.8 (*p* = 0.050) and 25.6 ± 5.7 vs. 21.9 ± 7.3 (*p* = 0.031), respectively. Further, a crossing over sign was identified to be more common in the nonCAM group (*p* = 0.091). In all cases, no pincer impingement was observed (Table 3).

For the rotational alignment, no significant differences were observed between groups; however, positive tendencies were found for the tibial plateau torsion with −4.9 ± 9.4 (CAM) vs. −9.0° ± 10.7° (nonCAM) (*p* = 0.084). All findings for the torsional alignments are presented in Table 4.

## 4. Discussion

Our results show that the pathology of CAM impingement in dysplastic and borderline dysplastic hips correlated significantly with the CE angle as well as the acetabular hip index. Positive tendencies were found for age and a lower BMI. On the other side, the rotational alignment of the lower extremity, especially of the femur and the acetabulum, did not affect the presence of CAM impingement in our cohort.

Although all patients who were included in this study presented with hip pain related to hip dysplasia, a combined FAI may exacerbate the symptoms. For definite radiographic diagnosis of FAI and DDH, a standardized X-ray, including an anteroposterior X-ray of the pelvis, an axial view and faux profile of the affected hip, is essential [16,17,18,19]. For diagnosis of intraarticular lesions, such as potential labral tears or chondral lesions, additional diagnostic tools, including rotational CT and MRI, are essential [20].

The current literature suggests that CAM impingement is found especially in young patients and typically associated with a countercoup lesion at the posterior inferior acetabular margin [21]. Biomechanically, larger total and anterior femoral head coverage in protrusio acetabula as well as acetabular retroversion or an extra-articular anterior supine hip impingement are described to be predisposing factors. Other correlating pathologies described in the literature include reduced femoral torsion [22] as well as an anterior iliac inferior spine [22,23]. For torsional alignment, a femoral retroversion (<5°) and increased femoral torsion (>35°) has been described to be associated with CAM impingement [24]. Another high prevalence described includes a combined femoral and tibial torsional abnormality with a mean femoral antetorsion of 23° and tibial antetorsion of 29° [25,26]. All these findings are thought to cause a mismatch between femoral head and acetabulum, leading to cartilage damage and, subsequently, to excessive growth of bone, the so-called os-acetabuli stress reaction, which can be identified as an asymmetry of the femoral head–neck junction [20]. For range of motion, limitations could be observed especially in hip flexion and internal rotation in the patients suffering from protrusio acetabula [1], whereas decreased femoral torsion showed less flexion and internal rotation in 90° flexion [22].

Wells J. et al. [27] investigated the proximal femoral characteristics and observed an incidence of cam deformity in 42%, which is comparable to the 36.5% in our cohort. Thus, the authors defined CAM-type FAI as an alpha angle greater than or equal to 55°; however, only 76% of patients tested positive for anterior impingement compared to 83% of patients with an alpha angle less than or equal to 55°. In the CAM-type FAI group, a reduced head–neck offset at the 1:30 point in 82% was noted, and a significant difference between mild and moderate-to-severe DDH was observed for femoral head–neck offset, respectively, with a femoral head–neck offset ratio of 12:00 (*p* = 0.04 and *p* = 0.01, respectively). For the femoral version in DDH, no significant correlation to the CAM-type FAI was observed [26]. In comparison to our trial, this study focused on the femoral characteristics and not the acetabular ones, except for the CE angle. Furthermore, no other torsional measurements were performed including acetabular torsion.

In our cohort, the acetabular hip index as well as CE were associated significantly with the development of CAM impingement. Thus, a low CE angle of 21.0° ± 5.4° (CAM) vs. 23.7° ± 5.8° (nonCAM) was significantly predisposing for impingement pathomorphology with a *p*-value of 0.05. For the acetabular hip index, a higher index of 25.6 ± 5.7 (CAM) compared to 21.9 ± 7.3 in the nonCAM group were observed (*p* = 0.031). This may be related to increased range of motion, especially of the abduction, causing chondral lesion at the head–neck junction. Another possible explanation could be the incomplete development of the hip in DDH, where the capsule is located more medial at the head–neck junction than in nonCAM hips. Subsequently, the acetabular hip index was higher avoiding any potential hip dislocation. Interestingly, a crossing over sign was identified to be more common in the nonCAM group, which is contradictory to the current literature. In DDH, a combination of retroversion may prevent too much range of motion and, therefore, prevent the development of CAM impingement. For femoral torsion, no significant differences were observed, although this was, overall, a bit higher than normal at 27.7° ± 12.6°. Likewise, with the torsional alignment, no correlation to the range of motion could be identified between the two groups.

For treatment, current guidelines for FAI suggest a symptomatic approach including acetabular trimming with hip arthroscopy, anterior inferior iliac spinal decompression and periacetabular osteotomies. The latter ones were discussed for reorientation, especially in retroverted hips with questionable outcome in the development of osteoarthritis [1], raising the question regarding which intrinsic factors may be involved in the development. These could either include genetic or epigenetic causes, such as collagen alpha-1(I) chain gene (COL1A1) and vitamin D receptor (VDR) [28,29], or an incomplete/inadequate chondrogenesis in DDH.

There were several limitations to this study. This study was of a retrospective design and only included 52 consecutive hips that were diagnosed with dysplastic and/or borderline dysplastic hips undergoing a rotational CT scan. Furthermore, to identify significant differences (*p* < 0.05) in the measurements between the CAM and nonCAM groups on the radiographs, small variations related to the positioning of the patient and the technique used to perform the X-rays could be observed. No long-term follow up was performed, which makes it difficult to discuss the development of osteoarthritis. Additionally, it was difficult to differentiate between the clinical symptoms resulting from the DDH abnormality or the FAI. Additionally, although no differences between DDH and borderline DDH were found, the subgroups were potentially underpowered. To minimize this error, all measurements of the plain radiography were performed by an orthopedic-surgeon trained observer and all measurements of the computed tomography by a specialized musculoskeletal trained radiologist. A consensus reading was not performed since the interobserver and intraobserver reliability were described as 0.911 and 0.955 for EOS, respectively, 0.934 and 0.934 for CT scan to measure the rotational alignment [30].

## 5. Conclusions

In dysplastic and borderline dysplastic hips, the AHI as well as the CE angle were significantly associated with the development of CAM impingement in our cohort. The crossing over sign was identified to be more common in the nonCAM group, which is contradictory to the current literature. These findings may suggest that in addition to the biomechanical abnormalities, intrinsic factors, including genetic and epigenetic causes, or incomplete chondrogenesis have an important role in the development of the FAI.

## Figures and Tables

**Table 1 jpm-12-01129-t001:** Demographics of the individuals presenting with symptomatic hip pain.

	Total	Coefficient	*p*-Value	CAM	nonCAM	*p*-Value
Numbers	52			19 (36.5%)	33 (63.5%)	
Female (%)	46 (88.5)	−0.139	0.520	16 (84.2)	30 (90.9)	0.238
Left hip (%)	25 (48.1)	−0.078	0.573	8 (42.1)	17 (51.5)	0.261
Age (years)	28.8 ± 7.6	0.014	0.134	30.9 ± 6.6	27.6 ± 7.9	0.067
Height (cm)	169.7 ± 8.4	0.013	0.822	171.6 ± 9.1	168.6 ± 7.9	0.125
Bodyweight (kg)	68.1 ± 12.2	−0.008	0.920	66.1 ± 9.7	69.4 ± 13.5	0.189
BMI (kg/m^2^)	23.8 ± 4.3	−0.003	0.990	22.4 ± 2.6	24.5 ± 5.0	0.059
Borderline DDH	13 (25.0%)	0.179	0.253	3 (15.8%)	10 (30.3%)	
DDH	39 (75.0%)	0.179	0.253	16 (84.2%)	23 (69.7%)	

**Table 2 jpm-12-01129-t002:** Hip range of motion in the CAM and nonCAM groups.

	Coefficient	*p*-Value	Total	CAM	nonCAM	*p*-Value
Flexion (°)	−0.011	0.120	130 ± 13.1	132.6 ± 12.8	128.5 ± 13.4	0.139
Extension (°)	−0.026	0.111	2.1 ± 4.1	2.6 ± 4.5	1.8 ± 3.9	0.249
External rotation (°)	0.009	0.202	52.8 ± 11.5	54.5 ± 11.4	51.8 ± 11.6	0.214
Internal rotation (°)	0.006	0.280	41.4 ± 14.3	41.3 ± 14.8	41.4 ± 14.2	0.495
Abduction (°)	0.010	0.139	51.8 ± 15.2	51.8 ± 15.9	51.8 ± 15.1	0.498
Adduction (°)	−0.007	0.467	26.9 ± 10.0	29.0 ± 11.5	25.8 ± 9.0	0.136

**Table 3 jpm-12-01129-t003:** Findings on the anteroposterior pelvis radiographs and axial views of the affected hips.

	Total	Coefficient	*p*-Value	CAM	nonCAM	*p*-Value
CE angle (°)	22.7 ± 5.8	0.002	0.968	21.0 ± 5.4	23.7 ± 5.8	0.050
AI angle (°)	11.2 ± 5.2	0.006	0.781	12.6 ± 6.3	10.3 ± 4.3	0.065
Sharp angle (°)	42.5 ± 3.7	−0.005	0.894	43.3 ± 3.5	42.1 ± 3.8	0.148
Hip lateralization index	0.56 ± 0.06	0.617	0.590	0.57 ± 0.06	0.56 ± 0.06	0.214
AHI	23.2 ± 7.0	0.017	0.524	25.6 ± 5.7	21.9 ± 7.3	0.031
CCD (°)	133.0 ± 5.7	<0.005	0.994	133.3 ± 6.4	132.8 ± 5.4	0.378
Crossing over sign	17 (32.7)	−0.194	0.184	4 (21.1)	13 (39.4)	0.091
Kellgren–Lawrence score	0.4 ± 0.5	0.015	0.909	0.4 ± 0.5	0.4 ± 0.5	0.488
Alpha angle (°)	100.0 ± 10.9	−0.001	0.922	101.5 ± 10.0	98.8 ± 11.6	0.235
Beta angle (°)	57.0 ± 7.5	0.015	0.288	58.8 ± 7.8	55.7 ± 7.2	0.111

**Table 4 jpm-12-01129-t004:** Rotational computed tomography findings.

	Total	Coefficient	*p*-Value	CAM	nonCAM	*p*-Value
Acetabular torsion (°)	18.9 ± 5.7	−0.005	0.712	18.6 ± 6.5	19.1 ± 5.3	0.373
Femoral neck torsion (°)	15.4 ± 10.7	0.012	0.516	15.7 ± 8.3	15.2 ± 12.0	0.432
Femoral condyle torsion (°)	−13.2 ± 9.8	−0.048	0.277	−12.4 ± 10.7	−13.7 ± 9.4	0.328
Femoral torsion (°)	27.7 ± 12.6	−0.014	0.469	26.5 ± 12.6	28.4 ± 12.7	0.305
Tibial plateau torsion (°)	−7.5 ± 10.4	0.042	0.305	−4.9 ± 9.4	−9.0 ± 10.7	0.084
Femorotibial torsion (°)	6.3 ± 5.0	−0.025	0.589	7.6 ± 6.5	5.5 ± 3.7	0.066
Ankle torsion (°)	28.9 ± 10.5	0.008	0.202	30.2 ± 11.2	27.9 ± 10.1	0.236
Tibial torsion (°)	37.0 ± 7.8	−0.008	0.336	36.6 ± 9.6	37.3 ± 6.5	0.377
Leg torsion (°)	−13.5 ± 14.0	−0.002	0.674	−14.5 ± 14.3	−12.8 ± 14.0	0.337

## Data Availability

Not applicable.

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
