# Peer review of "The Impact of Hip Dysplasia on CAM Impingement"

_jpm, 2022, doi:10.3390/jpm12071129_

Round 1

Reviewer 1 Report

Please see the attached file, thank you. 

Author Response

Thanks a lot. We included our multivariate analysis and adjusted the manuscript according to your suggestions. 

In contrast to the article you referenced to, our manuscript analyzes all standard measurements of the acetabulum and femur as well as the torsional alignment. 

Please find attached our revised manuscript

Reviewer 2 Report

The authors analyse the impact of dysplasia, quantifed though a number of X-rays-based parameters, on CAM impingment phenomenon.

My comments floow:

-I would suggest to integrate the Introduction section with explicative figures as well as with a wider discussion about impingment phenomenon and classification (not only CAM type exists).

-At the beginning of the Materials and Methods section you mention the CT, but it not clear to me how these images were then used. Please clarify.

-You identify the 'geometrical variables' most correlated with the occurrence of CAM impingment. Nevertheless, have you tried to assess if crosss-correlation existed between them? The calculation of the Variance Inflation Factor for example, might be a useful approach. Moreover, it would also be of interst if you could try to build a multivariate model taking all the significant varaibles into account.

Author Response

Dear Reviewer, thanks you so much. We included the findings of the multivariate analysis in the tables. In addition, the manuscript was adjusted accordingly and we included a detailed description of the CT measurements. 

Round 2

Reviewer 1 Report

Dear authors 

I congrats to you for the efforts in revising the manuscript.  Now the manuscript is an successful scientific report and should be accepted for further publication.